# Smartphone Addiction and Related Factors among Athletes

**DOI:** 10.3390/bs14040341

**Published:** 2024-04-18

**Authors:** Sultan Sahin Koybulan, Duygu Altin, Gorkem Yararbas, Hur Hassoy

**Affiliations:** 1Institute on Drug Abuse, Toxicology and Pharmaceutical Science, Ege University, Izmir 35040, Turkey; sultansahin33@gmail.com (S.S.K.); duygu.altin@gmail.com (D.A.); 2Department of Public Health, Faculty of Medicine, Ege University, Izmir 35040, Turkey; hur.hassoy@ege.edu.tr

**Keywords:** smartphone addiction, athletes, eating attitude, body perception

## Abstract

Smartphone addiction (SA) is increasing worldwide. The aim of this study is to determine the level of SA in athletes affiliated to the Turkish Handball Federation in Izmir and to examine its relationship with factors such as sociodemographic status, health status, eating attitude, and body perception. This cross-sectional study was conducted in March–April 2021 in Izmir Province. The sample of the study consisted of 212 licensed handball athletes. The short SA scale, three-factor nutrition scale, and body perception scale were used. A chi-square test was used for bivariate comparisons and logistic regression analysis was used for multivariate comparisons. The study was completed with 202 individuals (the coverage rate was 95.3%). The prevalence of SA was found to be 27.7%. The risk of SA increased 2.49-fold (CI: 1.17–5.31, *p* = 0.018) in female participants, 2.01-fold (CI: 1.01–4.06, *p* = 0.048) in participants with alcohol use, 2.17-fold (CI: 1.04–4.58, *p* = 0.042) in participants with low nutritional scores, 2.65-fold (CI: 1.15–6.10, *p* = 0.022) in individuals with high-income status, and 2.66-fold (CI: 1.07–6.64, *p* = 0.036) in individuals with high body perception scale score. In total, 27.7% of the athlete sample had scores above the SA threshold. These results point out that a behavioral addiction such as SA can occur even in professionals of an activity such as sports, known for beneficial effects in terms of healthy life.

## 1. Introduction

In today’s world, addiction includes behavioral addictions such as gambling, internet, shopping, eating, and smartphone addictions, in addition to substance use such as smoking, alcohol use, and drug use [1]. Behavioral addictions show psychological and physical symptoms (mood variability, mental preoccupation, interpersonal conflicts, tolerance, relapse, and withdrawal), which are the main components of addiction, as seen in alcohol and substance addictions [2].

Although the main purpose of the emergence of smartphone technology was to perform communication and reach information cheaply, easily, and safely, the rapid spread of use and pathological overuse has led to problematic internet use [3] and smartphone addiction (SA), which is a new type of addiction in the literature. SA can be defined as the inability to prevent the desire to use smartphones, excessive irritability and aggression in case of withdrawal, and the gradual negative impact and deterioration of the individual’s work, social, and family life [4]. Although it is not defined in DSM-5-TR or ICD-11, several studies show that SA is a very common addiction and its prevalence is gradually increasing [5,6].

When studies on SA are examined, it has been observed that pathological conditions that progress with repetitive and impulsive behavior patterns that stimulate the reward system are similar to the pathophysiology of other behavioral addictions and substance use disorders, and multiple neurotransmitter systems are involved in their mechanism [7]. In light of these studies, it is seen that similar dopaminergic and serotonergic systems are involved both in sports and eating [8,9]. In addition to the above-mentioned neurophysiological explanations for sport and eating, there are also scientific studies that contribute to psychological factors. Sport may increase the risk for eating disorders [10]. Athletes are exposed to sociocultural pressure to have an ideal body appearance [11]. It is also reported that psychological characteristics such as performance anxiety are risk factors for eating problems, and the onset of eating disorders coincides with the active-athlete age periods of adolescence and young adulthood [12].

Body perception, which is associated with the development of eating disorders [13] and has been evaluated as an important factor in an athlete sample [14], is defined as feelings, thoughts, and perceptions about one’s own body [15]. It is stated that negative body perception leads to social appearance anxiety [16], and individuals with social appearance anxiety use smartphones more than usual to cope with real-life situations [17]. Individuals with long-term smartphone use are exposed to more idealized body appearances (social media, blogs, advertisements, etc.), which may lead to cognitive internalization of the ideal body appearance, and this body comparison may result in negative body perception [18]. Sociocultural ideals about body appearance imposed through smartphone use are particularly high in the athlete population and may cause the difference between actual appearance and perceived appearance to grow, thus resulting in negative body perception [14].

Sports are generally known to have beneficial effects on both prevention and rehabilitation processes from addictions [19]. However, there are studies showing a positive association between physical activity and types of problematic use of the internet [20]. These contradictory results need more clarification. Recent research findings demonstrate that smartphone addiction can escalate to levels as high as 40% among athletes in Singapore [21] and sports science students in Turkey [22]. Additionally, these studies delve into its correlations with various factors within athlete samples in Turkey [23]. Handball is evaluated as a sport with high levels of physical and mental stress [24], which are mediating factors for eating and internet addiction [25]. Being a team and competitive sport, requiring high motivation, a sense of achievement depending on the struggle, and a high level of physical contact are handball-specific features that can support the development of impulses for dopamine release in players [26]. The unique aspect of this study is that it examines the risk of SA in a population with intensive sporting activities in their lives and questions the associated factors. The main question of the study is the relationship of SA, important in terms of reward pathways in the athlete population, where neurotransmitters such as dopamine and serotonin are intensively activated, with eating status and body perception.

This study aimed to examine the levels of SA and the factors that may be related to SA in Turkish professional athletes in Izmir, affiliated to the Handball Federation. In this context, the study focused on the relationship between athletes’ SAs and their socioeconomic and demographic characteristics, existing health status and habits, body perception levels and eating attitudes.

## 2. Materials and Methods

### 2.1. Research Design

This research was an analytical study planned with a cross-sectional design. It was conducted in the districts where there are sports clubs with licensed athletes in the handball branch in the province of Izmir, Turkey, in the 2019–2020 and 2020–2021 seasons (Appendix A).

### 2.2. Study Sample

The population of the study consisted of athletes born before 2007 who were licensed and active in sports clubs in Izmir Province, registered with the Handball Federation in the 2019–2020 and 2020–2021 seasons. Athletes playing in the Professional League and Youth League athletes who had the competence to play in the Professional League were included in the study. The number of licensed athletes (league athletes and licensed youth team athletes) affiliated to the Handball Federation of Izmir Province is 212. The athletes were contacted by the federation’s communication department and the sports clubs to which they were affiliated. No sample was selected in the study; we aimed to include the entire population of 212 people. All athletes who gave consent to participate were included in the study.

Approval was obtained from the Ege University Medical Research Ethics Committee for Research in February 2021 (approval number: E.43687). Necessary permissions were obtained from the Turkish Handball Federation. The study did not involve any animal or human experimentation and consisted entirely of data collection by questionnaire and an analysis of the results.

### 2.3. Procedure

The data collection process of the research took place between March and April in 2021, amidst the ongoing COVID-19 pandemic. Sports clubs that requested participation were visited during their training hours for face-to-face data collection. For sports clubs that preferred not to engage in face-to-face meetings due to the pandemic, data were gathered remotely through Google Form surveys. The data were gathered directly by the researcher without the involvement of surveyors or interviewers.

### 2.4. Variables

The dependent variable of the study was SA. The 10-question “SA Scale-Short Form (SA-SF)” developed by Kwon et al. was used to measure the SA variable. The scale is evaluated on a 6-point scale. Each item is scored from 1 to 6, and the total score is between 10 and 60. As the scores obtained by the individual increases, the risk of addiction increases. When an ROC analysis of the scale is performed, the cut-off point is 33 for women and 31 for men. The Cronbach alpha value of the scale was found to be 0.97 [27]. Turkish validity and reliability studies were completed by Noyan et al. in 2015. The scale is single-factor and has no subscales [4]. The Cronbach alpha coefficient of the Turkish scale consisting of ten items was 0.867. Thus, the scale was accepted as a valid and reliable scale to examine SA in young adults.

The independent variables were sociodemographic characteristics, current health status and behaviors, nutrition behavior/attitude, and body perception. Sociodemographic characteristics such as the age, gender, income level (those with a monthly income of TRY 750 or less were considered low and those with an income of more than TRY 750 were considered high), and education level of the participants were evaluated. On 31 March 2021, in the data collection period, USD 1 equaled TRY 8.32. In order to comprehensively integrate the participant’s age into the study, an open-ended inquiry regarding their date of birth was employed. Participants’ current body weight and height were initially gathered in an open-ended format, with a subsequent computation of Body Mass Index (BMI). Regarding the existing health status and behaviors variable, there were questions describing their habits and existing conditions, such as whether they had past diagnosed diseases, the number/hours of weekly training, and smoking/alcohol use. In this section, athletes were asked to answer a questionnaire consisting of open and closed-ended questions. For the assessment of the eating behavior and attitude variable, the short version of the 51-question three-factor eating questionnaire (TFEQ), developed by Messic and Stunkard in 1985, was used to examine and measure the behavioral and cognitive components of nutritional status [28]. Cronbach’s alpha values of the cognitive restraint, uncontrolled eating, and emotional eating subscales are 0.76, 0.83, and 0.85, respectively [29]. Validity and reliability tests of the TFEQ in Turkish were conducted by Kirac et al. in 2015. The questionnaire consists of 18 questions and measures the degree of cognitive restraint of eating, the level of uncontrolled eating, and the degree of eating when one is emotional. It has also been reported to measure the level of sensitivity to hunger [30].

In the study, data on athletes’ satisfaction with their bodies were measured with the body perception scale. The scale, whose original name is the body cathexis scale (BCS), was first developed by Secord and Jourard in 1953 [31]. The reliability of the test for body image perception was determined to be r = 0.81. The body perception part of this scale developed by Secord and Jourand was translated into Turkish by Hovardaoğlu in 1989, and Turkish validity and reliability studies of the body perception scale were conducted by Hovardaoğlu in 1992 [32]. The scale consists of 40 items evaluating satisfaction with a part or function of the body. The scale is a 5-point Likert scale, and the total score can vary from a minimum of 40 to a maximum of 200 points. The scale includes the following: 1 = I like it a lot, 2 = I like it, 3 = I am undecided, 4 = I do not like it very much, and 5 = I do not like it at all, and those with high scores are defined as having low body perception. An increase in the total score indicates a decrease in satisfaction with body parts or functions, whereas a decrease in the total score indicates an increase in satisfaction. The cut-off point of the scale is 135 [33].

### 2.5. Statistical Analysis

Number, percentage, mean, minimum–maximum values, and standard deviation (SD) were used for the descriptive results of the study group. A chi-square test was used to evaluate the relationship between the independent variables and SA and for bivariate analysis. Effect sizes were calculated for establishing the differences between addicts and non-addicts in relation to the independent variables. The Φ (phi) coefficient was used for 2 × 2 contingency tables and Cramér’s V was used for the tables larger than 2 × 2 for the estimation of effect size [34]. Logistic regression analysis was used for multivariate comparisons. The dependent variable of the study was binary, and the observations were independent. The independent variables included in the regression model were not correlated with each other. The sample size required was sufficient. Therefore, the assumptions for performing logistic regression analysis were met. The significance level was taken as *p* < 0.05. IBM SPSS Statistics for Windows, version 25.0 (IBM Corp., Armonk, NY, USA), was used for data analysis.

## 3. Results

From the group of 212 people included in the research, 202 people were interviewed. Accordingly, the coverage rate of the study was determined as 95.3%. The SA status of the study group according to the sociodemographic characteristics and health data is presented in Table 1.

A total of 84.1% of the participants were high school students and under the age of 23, and 62.8% of the participants were female. When the educational status of the participants was analyzed, it was found that 67.8% were middle/high school graduates and 32.2% were university graduates. When the amount of income per capita was analyzed, it was found that 28.7% of the participants stated that they had a low-income level. The BMI of 79.2% of the participants was between 18.50 and 24.99, which is the ideal level. A total of 89.1% of the participants did not have any disease, and 17.8% of them smoked at least one cigarette a day. While 67.8% of individuals did not consume alcohol at all, 32.2% consumed alcohol at different frequencies, including rare consumption. It was also noted that all participants (100%) used smartphones.

In our study, SA was found to be higher in females than males, indicating a significant association of small effect size (*p* = 0.014, phi = 0.178). In the study group, those with a higher per-capita monthly income were found to have a higher level of phone addiction with a small effect size (*p* = 0.038, phi = 0.149). There was no statistically significant relationship between the participants’ age, educational status, financial status perception, and the occurrence of SA (*p* > 0.050).

While SA was found to be higher in individuals with alcohol use (*p* = 0.020, phi = 0.165), SA was found to be lower in individuals who actively trained/ participated in sport (*p* = 0.041, phi = 0.144), indicating small effect sizes for both associations. There was no statistically significant relationship between the variables of BMI, medication use, the presence of a diagnosed disease, and smoking and SA (*p* > 0.050).

The distribution of the scores obtained by the research group from the SA scale is presented in Figure 1.

According to the data, 28.2% of the participants reported that they disrupted their planned work because of their smartphone, 32.7% reported that they could not complete their lessons and work because of their smartphone, 29.3% reported that they thought about their smartphone even if they did not use it, and 26.8% reported that they insisted on using the smartphone despite it disrupting their lives (Figure 1).

Body perception values were 87.2 ± 29.4 on average, with a minimum of 40 and a maximum of 192. According to nutritional data, individuals consume an average of 2.58 ± 0.70 meals per day, with a minimum of 1 and a maximum of 5 main meals. They consume an average of 1.86 ± 1.07 snacks, with a minimum of 0 and a maximum of 6 snacks. The mean total score of the three-factor nutrition scale was 41.89 ± 11.2, and the mean subscale scores of the athletes were 11.84 ± 3.69 for uncontrolled eating/inability to restrain eating, 6.61 ± 2.94 for emotional eating, 15.04 ± 4.21 for conscious restriction, and 8.39 ± 3.82 for hunger (Appendix A).

SA was found to be lower in individuals with adequately balanced nutrition (*p* = 0.002). SA was statistically higher in individuals with higher scores on the nutritional status scale (*p* = 0.017, Cramer’s V = 0.244), on the uncontrolled eating subscale (*p* = 0.002, Cramer’s V = 0.244), and on the emotional eating subscale (*p* = 0.001, Cramer’s V = 0.290), with a medium effect size, whereas on the hunger subscale, a small effect size was observed (*p* = 0.030, Cramer’s V = 0.184). No significant relationship was found between the inability to restrict eating subfactor and SA (*p* > 0.050).

When body perception and SA were examined in the study group, liking one’s own body structure and body perception scale score were found to be statistically significant with SA. SA was found to be lower in individuals who stated that they liked their own body structure (*p* = 0.019). SA was found to be higher in individuals with higher body perception scale scores (*p*= 0.030) (Table 2).

The regression analysis results of the variables associated with SA in the research group are presented in Table 3.

In order to examine the variables that were found to be statistically significant as a result of binary analyses in detail, the variables associated with SA were evaluated by regression analysis. When the factors affecting SA were examined, statistical significance was found in the regression analysis between SA and gender, alcohol use, three-factor nutrition scale score, income status, and body perception scale score variables (*p* < 0.050).

In our study, the risk of SA increased 2.49-fold (CI: 1.17–5.31, *p* = 0.018) in female participants, 2.01-fold (CI: 1.01–4.06, *p* = 0.048) in participants with alcohol use, 2.17-fold (CI: 1.04–4.58, *p* = 0.042), 2.65-fold (CI: 1.15–6.10, *p* = 0.022) in individuals with high-income status, and 2.66-fold (CI: 1.07–6.64, *p* = 0.036) in individuals with high body perception scale score. In the regression analysis, the variable of doing active sports lost its significance for SA (*p* > 0.050).

## 4. Discussion

This study was conducted in the districts where there are sports clubs with licensed athletes in the Handball Branch in the province of Izmir in the 2019–2020 and 2020–2021 seasons. The study aims to examine the relationship between SA and factors such as socioeconomic status, health status, eating attitude, and body perception that may be related to SA in athletes residing in Izmir Province affiliated to the Handball Federation. Within the scope of the study, 202 handball athletes were interviewed.

As a result of the analysis, the frequency of SA of the participants was found to be 27.7%, and it was seen that all four hypotheses were confirmed in terms of certain variables. The risk of SA was higher in women, in individuals with high income, in participants with alcohol use, in individuals with low nutrition scale scores, and in individuals with low body perception scale scores.

### 4.1. Strengths and Limitations

Since this study was conducted among the athletes affiliated to the Handball Federation in Izmir Province, its generalizability is limited. Since the research was planned cross-sectionally, the cause-and-effect relationship was examined at the same time, so its causality and examination power are limited. The answers given to the questions in the questionnaire were limited to the accuracy of the answers given by the subjects. As a limitation specific to SA-SF, while it provides an effective means to predict smartphone addiction tendencies, it does not diagnose clinical addiction due to not being performed in clinical settings [27]. Regarding the body perception scale, it primarily focuses on perceptual aspects of body image, possibly not capturing other important dimensions such as the emotional impact of body image concerns. Cultural variabilities in body ideals, norms, and attitudes may also cause validity and reliability concerns in different cultural contexts [35]. The TFEQ, being a Likert-type scale to assess eating behaviors, may not fully capture the complexity and variability of individuals’ eating patterns and motivations, which may also be a limitation for the abovementioned two scales utilized in the research as well [36]. When the study is examined in terms of its originality, it is one of the first studies to examine SA, which is a behavioral addiction, as well factors such as socioeconomic status, health status, eating attitude, and body perception that may be related to it, in an athlete sample. In our study, it was observed that the female population was higher than the examples in the literature. This higher proportion of women is due to the lower number of male handball players in Izmir Province.

The frequency of players’ participation within the team, whether as regular players or as substitutes, as well as the success or failure of their team, may influence addictive behaviors. However, it is important to note that we did not inquire about these factors, representing another limitation of our study.

In the research design, it was planned to collect the data face to face. However, the working period coincided with the COVID-19 pandemic period. Therefore, 15% of data were collected online from participants. While there may be biases stemming from different data collection methods, it is believed that these differences did not significantly affect our results. Apart from the data collection procedure, the pandemic may be a factor affecting behavioral addictions as a coping mechanism for mental distress prevention [3,37,38]. The onset of the COVID-19 pandemic furnished an unprecedented opportunity for researchers worldwide to delve into the effects of stressful life events on individuals’ psychological reactions and addictive inclinations. During the COVID-19 pandemic, lockdown measures compelled individuals to spend extensive periods in front of screens, particularly smartphones, to sustain social connections, communication, and fulfill work and educational obligations. Hence, it becomes imperative to grasp the gravity of these addictive behaviors and their interplay with other pandemic-induced factors [37].

### 4.2. Comparison of the Results with the Literature

Among the socioeconomic variables, gender and average monthly income variables were found to be associated with SA. According to the data obtained in the study, SA risk was high in women. When the literature data are examined, gender, especially being male, is reported to be risky for pathological internet use [39]. In contrast to internet addiction and use, in SA studies, women were found to have a higher risk of addiction than men [40]. When the literature is evaluated in terms of the basis of this difference, it has been reported that the purpose of men’s smartphone use may be different from women’s; men use smartphones more as a tool, while women use them for the continuity of communication with their loved ones, and therefore, women have higher addiction levels [41]. In another study in which the difference in the purpose of use was reported, it was stated that women mostly used their phones for social media purposes, while men used their phones more intensively for gaming purposes [42]. In this context, studies in the literature have frequently found that women have a high level of SA. Another situation that supports the high prevalence of SA in women is that women have higher levels of self-awareness about their addictions and are more open about explaining their problems [27]. Our results are in line with the literature in terms of indicating that women are more addicted to smartphones.

#### 4.2.1. Socio-Demographic Variables and Prevalence of Smartphone Addiction

The prevalence of SA of the participants was found to be high. This high prevalence of SA in handball players can be explained in different ways. Firstly, behaviors with addictive potential such as exercise, eating, and internet use have common characteristics with addictions caused by substances in terms of affecting brain reward systems. The brain stimulation required for the reward (reward threshold) increases with the repetition of the addictive behavior [43]. Considering the professional athlete group that constituted the sample of the study, it can be said that they have highly activated reward systems due to their regular and intensive training programs. In our study, it is thought that the athletes choose SA as an easily accessible source of dopamine that would not harm their sportive performance during non-sporting hours as a result of the increase in the reward threshold caused by long-term sports. Another explanation for this high prevalence could be associated with the exacerbating effect of COVID-19 on SA [3]. The lockdown and social distance measures taken during the COVID-19 pandemic forced people to use their smartphone as a means of secure communication and access to information.

Another important finding of the study is that the risk of SA is higher in individuals with high-income status. According to a study conducted on middle school students in South Korea, high-income status is associated with intensive smartphone use [44]. The literature and study data on income status are in parallel. The main reason for this similarity may be related to the fact that individuals with high-income status have more access and tendency to use smartphones [45]. Financial affordability is cited as another factor contributing to the association between smartphone addiction and a higher income level [46].

#### 4.2.2. Alcohol Use and Smartphone Addiction

In this study, it was found that SA was 2 times more common in those who consumed alcohol. Studies have shown that there is a relationship between SA and alcohol use [47]. There are studies suggesting that this significant relationship can be explained by a potential psychosocial process that may contribute to behavioral addictions such as anxiety, depression, and neuroticism [48,49]. It is stated that alcohol use in athletes is stress-related. Especially in team games such as handball, alcohol use was found to be high [24]. Alcohol use, similar to SA, is reported to occur for the control of internal dysphoric states [50]. The COVID-19 pandemic, which aligns with the data collection period of this research, could serve as another factor exacerbating both alcohol consumption and smartphone addiction, thereby bolstering their correlation. This pandemic is notorious for instigating stress as a result of lockdown measures, uncertainties, social isolation, and grave health risks. Research has shown increases in the number of people with alcohol use problems [51] and smartphone addiction [52] during the pandemic. Alcohol and smartphone use can be a coping mechanism for pandemic-specific conditions. Sport increases the level of reward components (dopamine and endogenous opioids). However, when reward components are not activated, athletes turn to alcohol, which results in an increase in these secretions [53]. There are studies suggesting that alterations in this reward system may underlie substance and behavioral addictions [54]. Studies showing that “reward deficiency syndrome” resulting from such changes in dopaminergic receptor levels leads to substance seeking and other behavioral patterns [7] are also supported by findings on genetic determinants [55]. In light of these studies, it can be said that the relationship between SA and alcohol use in handball players may be related to reward deficiency syndrome and may be aimed at coping with internal dysphoric moods and stress, which are seen at higher levels in team sports.

#### 4.2.3. Eating Behavior and Smartphone Addiction

In this study, individuals with a high mean total score on the three-factor nutrition scale (TFEQ) were found to be at higher risk for SA. In studies conducted with different athletes and student groups, including handball, total TFEQ scores were found to be associated with SA [56,57]. When different studies were examined, TFEQ scores and subscale scores were found to be associated with SA, in line with the literature. It is thought that this relationship between problematic eating behavior and SA can be explained by common pathways involved in brain reward systems [58]. Another explanation for the association between eating and internet addiction is the mediating effect of psychological distress [25]. Emotional eating often arises due to heightened levels of the stress hormone cortisol, which subsequently amplifies both our appetite and inclination for food consumption. Engaging in stress-induced eating, particularly amidst the onset of an infectious disease outbreak, is a prevalent occurrence [59]. Moreover, both eating and smartphone usage can serve as a means of diversion or relief from the isolation imposed by the pandemic. There are also studies showing that athletes with eating disorders have a higher need for social approval [60]. Problematic smartphone use is associated with the need for social approval [48]. These data suggest that the need for social approval may mediate the relationship between SA and eating scale scores in athletes.

#### 4.2.4. Body Perception and Smartphone Addiction

In this study, a high mean body perception scale score for athletes was found to be riskier in terms of SA. In a study conducted on university students, similarly, individuals with negative body perception were found to have high levels of SA [61]. It is thought that the basis of this relationship may be the pathological desire to be liked, changing popular understanding of beauty, and social perceptions that increase with developing technology and smartphone use. Individuals with a negative body perception may have low self-esteem in real life and may need to receive more positive reactions through smartphones. Individuals with low self-esteem and limited social skills, driven by a strong need for social approval and a sense of belonging, are more predisposed to developing a smartphone addiction [62]. Social networks provide individuals with an opportunity to show only their positive characteristics and can facilitate them to create the perception they desire [63]. Body perception is another factor shown to be affected by the COVID-19 pandemic. The pandemic has ushered in shifts in lifestyles [38], notably characterized by heightened sedentary behaviors such as extended periods spent at home, which have been linked to a generally unfavorable perception of body image [64].

## 5. Conclusions

In this study conducted on handball athletes, the prevalence of SA was found to be 27.7%. The risk of SA was found to be higher in women, those with higher income levels, those with alcohol use, those with higher eating scores, and those with negative body perception. These results regarding the comorbidity of SA with eating problems, alcohol use, and negative body perception indicate the presence of a reward system disorder. This study contains striking results in terms of pointing out that addiction is not a problem that affects only certain segments of society and that a behavioral addiction such as SA can occur even in professionals of an activity such as sports, which is known to have beneficial effects on health. There is a need for interventions that will allow those working in the field of guidance and counseling to approach the risk groups defined above with special care and to create healthy resources that individuals can use to stimulate their reward systems.

## Figures and Tables

**Figure 1 behavsci-14-00341-f001:**
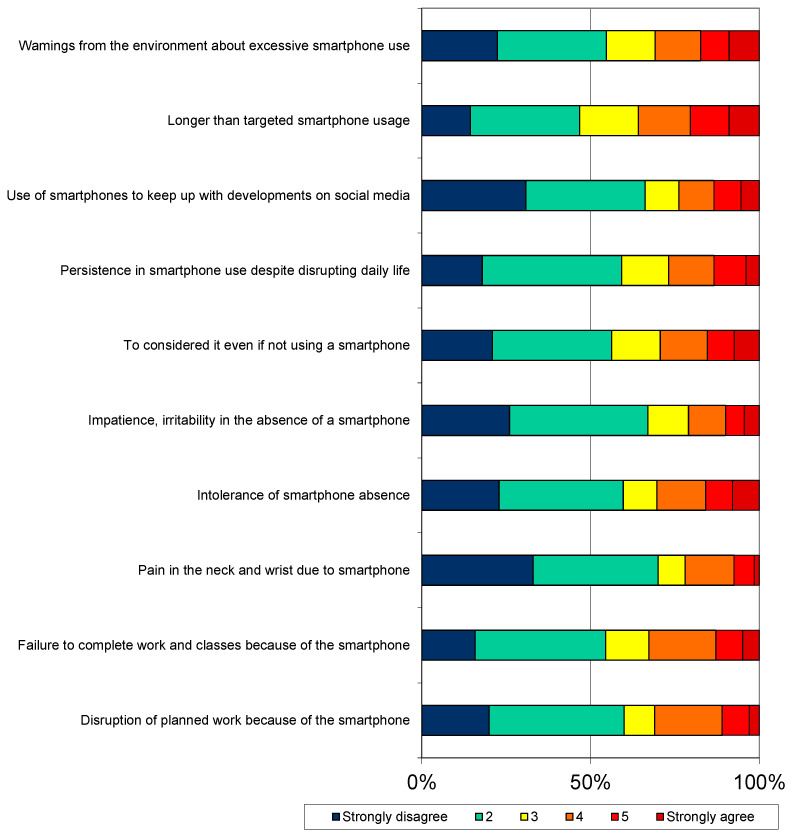
Distribution of the scores obtained by the research group from the SA scale.

**Table 1 behavsci-14-00341-t001:** SA status of the study group according to socioeconomic, demographic, and health data.

Sociodemographic and Economic Characteristics		Addicted to Smartphones		Not Addicted to Smartphones		Total **		Effect Size	*p*
		N	%	N	%	N	%		
Age	23 years and under	48	28.2	122	71.8	170	84.1	0.026	0.708
	24 years and older	8	25.0	24	75.0	32	15.9		
Sex	Female	43	33.9	84	66.1	127	62.9	0.178	0.011 *
	Male	13	17.3	62	82.7	75	37.1		
Education Status	Middle school–high school	38	27.7	99	72.3	137	67.8	0.000	0.995
	University and above	18	27.7	47	72.3	65	32.2		
Monthly Average Income	Low	10	17.2	48	82.8	58	28.7	0.149	0.035 *
	High	46	31.9	98	68.1	144	71.3		
Active Training/Sports	Yes	50	26.2	141	73.8	191	94.6	0.144	0.041 *
	No	6	54.5	5	45.5	11	5.4		
Diagnosed Disease	Yes	7	31.8	15	68.2	22	10.9	0.032	0.649
	No	49	27.2	131	72.8	180	89.1		
BMI Groups	17.00–18.49	1	20.0	4	80.0	5	2.5	0.034	0.887
	18.50–24.90	44	27.5	116	72.5	160	79.2		
	25.00 and above	11	29.7	26	70.3	37	18.3		
Cigarette Use	I don’t smoke	43	25.9	123	74.1	166	82.2	0.087	0.215
	Smoking	13	36.1	23	63.9	36	17.8		
Alcohol Usage	Yes	25	38.5	40	61.5	65	32.2	0.165	0.019 *
	No	31	22.6	106	77,4	137	67.8		

*p* < 0.05 for the data indicated with *. ** Percentage of columns used.

**Table 2 behavsci-14-00341-t002:** SA status of the study group according to nutrition and body perception data.

Nutrition Specifications		Addicted to Smartphone		Not Addicted to Smartphone		Total **		Effect Size	*p*
		n	%	n	%	n	%		
Body perception score	48–134 points	48	25.5	140	74.5	188	93.6	0.179	0.025 *
	135–192 points	8	57.1	6	42.9	14	6.4		
Three-factor nutrition score	20–29	3	10.0	27	90.0	30	14.8	0.244	0.017 *
	30–39	11	20.0	44	80.0	55	27.1		
	40–49	28	38.4	45	61.6	73	36.1		
	50–59	7	25.9	20	74.1	27	13.4		
	60–69	7	41.2	10	58.8	17	8.4		
Uncontrolled eating factor	5–9 points	7	12.1	51	87.9	58	28.7	0.244	0.002 *
	10–14 points	29	30.2	67	69.8	96	47.5		
	15+ points	20	41.7	28	58.3	48	23.8		
Emotional eating factor	3–6 points	17	15.7	91	84.3	108	53.5	0.290	<0.001 *
	7–10 points	27	39.7	41	60.3	68	33.7		
	11 + points	12	46.2	14	53.9	26	12.8		
Inability to constrain factor	4–7 points	11	24.4	34	75.6	45	22.3	0.075	0.567
	8–11 points	30	31.3	66	68.8	96	47.5		
	12+ points	15	24.6	46	75.4	61	30.2		
Hunger factor	4–7 points	16	18.4	71	81.6	87	43.0	0.184	0.033 *
	8–11 points	23	33.3	46	66.7	69	34.2		
	12+ points	17	37.0	29	63.0	46	22.8		

* *p* < 0.05. ** Percentage of columns used.

**Table 3 behavsci-14-00341-t003:** Regression analysis results of the variables associated with SA.

Variables		OR	(%95 GA)	*p*
Sex	Ref: Male	1		
	Female	2.49	GA: 1.17–5.31	0.018 *
Alcohol use	Ref: No alcohol consumption	1		
	Alcohol consumption	2.01	GA: 1.01–4.06	0.048 *
Active training/sports	Ref: Regular sports are practiced	1		
	Lack of regular exercise	2.04	GA: 0.50–8.39	0.320
Three-factor nutrition score	Ref: Low-scale score	1		
	High scale score	2.17	GA: 1.04–4.58	0.042 *
Income	Ref: Low income	1		
	High income level	2.65	GA: 1.15–6.10	0.022 *
Body perception score	Ref: Low scale score	1		
	High scale score	2.66	GA: 1.07–6.64	0.036 *

* *p* < 0.050.

## Data Availability

Data are contained within the article and Appendix A.

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
