# Peer review of "Smartphone Addiction and Related Factors among Athletes"

_behavsci, 2024, doi:10.3390/bs14040341_

Round 1
Reviewer 1 Report
Comments and Suggestions for Authors
Thank you for the opportunity to review this interesting and current study.
However, I think it can be improved.
It appears that there was an effort to have recent bibliographic references, however, sometimes when you want to characterize the current state, they use references that are more than 10 years old. For example, in the Introduction: “In today's world, addiction includes behavioral addictions such as gambling, internet, shopping, eating and smartphone addictions in addition to substance use such as smoking, alcohol and drugs (M. Kwon et al., 2013).
In other words, to characterize today's world it uses a bibliographic reference that is 11 years old. There are much more recent references.
The way data was collected in each club and online, the training of interviewers, etc., should be better specified.
Regarding the limitations of the study, I think they should be more specified, namely: - those inherent to the instruments used and the method of collection; and essentially to the fact of the existence of the Covid 19 Pademia.
It is true that they refer to the importance that the pandemic may have had on the results (pp 284-287 “Another explanation for this high prevalence could be associated with the exacerbating effect of COVID-19 on SA (Kamolthip et al., 2022). The lock down and social distance measures taken during COVID-19 pandemic forced people to use smartphone as a means of secure communication and access to information”), however, this is later not reflected in the discussion.
I think they could have more bibliographical references about the pandemic and integrate this into the discussion itself. It would also be interesting to know more about the players' sporting performance. If they played regularly in the team or were substitutes, if their team was successful or unsuccessful, etc. Could this not influence addictive behaviors? If they have not collected this data, they should mention this in the limitations and take this into account in the discussion.
In fact, they recognize this when they mention: pp. 321-325 “There are also studies showing that athletes with eating disorders have a higher need for social approval (Langbein et al., 2021). Problematic smartphone use is associated with the need for social approval (Gutiérrez et al., 2016). These data suggest that the need for social approval may mediate the relationship between SA and eating scale scores in athletes. However, they later did not participate in the discussion.”
Reviewer 2 Report
Comments and Suggestions for Authors
The evaluated manuscript focuses on examining the relationship between smartphone addiction and sociodemographic characteristics, current health status and behaviors, nutritional behavior/attitude, and body perception in Turkish athletes. The subject matter of the article is highly pertinent, and the findings obtained have the potential to offer valuable insights for the formulation and implementation of prevention policies.
Overall, the introduction is accurately written, well-structured, and offers a concise summary of the variables being studied. However, I believe the introduction could benefit from considering the following aspects:
Although the term "smartphone addiction" is widely used in scientific literature and the authors adequately justify its usage, it would be advisable to clarify that it is not a recognized addictive disorder in either the DSM-V or the ICD-11.
Since the authors aim to establish the level or prevalence of smartphone addiction among professional athletes, it would be advisable to discuss prevalence rates obtained in previous studies in the introduction, both globally and in equivalent samples.
It would also be advisable to establish in the introduction whether there are similar studies examining the level and associated factors in athletes or professional sportspeople both in Turkey and in other countries.
In relation to the method, the following should be considered:
Assessing the reliability of the tests for the sample, using measures like Cronbach's alpha or McDonald's omega, would be advantageous for ensuring the consistency and accuracy of the measurements, since the SAS-SF seems to reference the internal consistency of the scale validation study in Turkish, while in the rest of the scale, no reliability data is provided. Additionally, including an example item from each test used could help illustrate the nature of the assessments.
Regarding the health status and behaviors questionnaire, it is mentioned that participants answered both open-ended and closed-ended questions. It would be advisable to explain this point, especially regarding which variables were addressed with open-ended questions and how they were coded.
Regarding the results, an important gap is the lack of reporting effect sizes when establishing the differences between addicts and non-addicts, both in relation to demographic variables and in relation to the results of other standardized tests. Through these effect sizes, readers could appreciate the magnitude of the differences found.
No information is provided regarding the fulfillment of assumptions for conducting a logistic regression analysis.
Overall, the discussion is quite comprehensive, although there is one issue I would like to see addressed:
It is mentioned that individuals with higher incomes have a greater tendency to suffer from smartphone addiction, but the reason is not discussed.
Reviewer 3 Report
Comments and Suggestions for Authors
The study examined smartphone addiction (SA) among Turkish handball athletes and its relation to various factors. The prevalence of addiction was 27.7%. Risk factors included being female, alcohol use, low nutritional scores, high income, and high body perception scale score. The study suggests that even athletes can be prone to such behavioral addictions.
General Feedback:
1. The introduction appears to be a bit long and could benefit from a more focused discussion on the prevalence of smartphone addiction among athletes, referencing relevant studies where possible.
2. The tables presented in the paper are well-structured and informative.
3. The discussion section, while comprehensive, could be more succinct and better organized with the inclusion of additional subsections.
Specific Feedback:
- Please verify the accuracy of the affiliation "Ege University" and provide the complete address as per the author instructions.
- On line 10, the phrase "This cross-sectional study conducted in March-April 2021 in Izmir province" should be revised to "This cross-sectional study was conducted in March-April 2021 in Izmir province."
- Please ensure that the in-text citation style aligns with the journal's requirements.
- On lines 78/81, consider starting a new paragraph when introducing the research question and the aim of the study.
- On line 86, revise the section title to "Materials and Methods" as per the journal's guidelines. Also, avoid capitalizing all the text in the section titles.
- Please provide a full citation for the Kwon et al. study on line 106.
- Citations for the BCS on line 133, and for HovardaoÄŸlu on lines 134 and 136, should be included.
- On line 145, please add the acronym "SD" for "standard deviation."
- Lastly, Figure 1 could be improved with higher quality graphics. Please consider revising it.
Comments on the Quality of English LanguageThe manuscript would benefit from further proofreading to correct occasional typographical errors and missing words.
Round 2
Reviewer 1 Report
Comments and Suggestions for Authors
We consider that the changes introduced by the authors improve the article and essentially respond to what was suggested. In our opinion, the article meets the conditions to be published.
Author Response
Thank you very much for your valuable contributions.
Reviewer 2 Report
Comments and Suggestions for Authors
The only point I would like you to modify is that in relation to the TFEQ, you indicate the reliability of the three subscales separately instead of the range.
Author Response
|
Comments 1: The only point I would like you to modify is that in relation to the TFEQ, you indicate the reliability of the three subscales separately instead of the range. |
|
Response 1: Thank you for your invaluable feedback. The Cronbach’s alpha value for the subscales TFEQ are added separately in line 136 and 137.
|

Reviewer 3 Report
Comments and Suggestions for Authors
1. Line 43: Please correct the reference from "DSM-V" to "DSM-5". Additionally, it would be beneficial to cite the most recent version, which is the "DSM-5-TR".
2. Line 137: Kindly reverse the order of the acronym and the full term. The correct format should be "Body Mass Index (BMI)".
3. Captions of Tables and Figures: Please adhere to the standard capitalization rules in the captions of tables and figures. Avoid capitalizing every word.
4. Line 275: There appears to be an in-text citation formatted in APA style. Please revise it to align with the citation guidelines of this journal.
5. Headers in Lines 326, 350, 378, and 397: Please ensure that the headers adhere to the correct style as per the journal's guidelines.
Author Response
|
Comments 1: Please correct the reference from "DSM-V" to "DSM-5". Additionally, it would be beneficial to cite the most recent version, which is the "DSM-5-TR". |
|
Response 1: Your constructive feedback is greatly appreciated. Required revision is performed accordingly. |
|
Comments 2: Line 137: Kindly reverse the order of the acronym and the full term. The correct format should be "Body Mass Index (BMI)". |
|
Response 2: Thank you for your invaluable feedback. The revision is performed.
Comments 3: Captions of Tables and Figures: Please adhere to the standard capitalization rules in the captions of tables and figures. Avoid capitalizing every word. Response 3: Thank you for your insightful feedback. The style has been meticulously revised in alignment with the provided guidelines.
Comments 4: Line 275: There appears to be an in-text citation formatted in APA style. Please revise it to align with the citation guidelines of this journal. Response 4: Thank you for your valuable feedback. Revision is performed.
Comments 5: Headers in Lines 326, 350, 378, and 397: Please ensure that the headers adhere to the correct style as per the journal's guidelines. Response 5: Thank you for your insightful feedback. The style has been revised in line with the provided guidelines.
|
